# LEARNING FINE-GRAINED PARAMETER SHARING VIA SPARSE TENSOR DECOMPOSITION

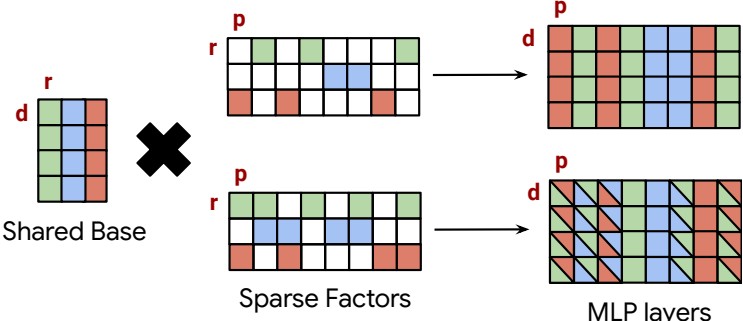

Figure 1: **Fi**ne-grained **P**arameter **S**haring (FiPS).

## ABSTRACT

Large neural networks attain cutting-edge performance on many tasks, yet their sheer size hinders deployment on resource-constrained devices. Among existing compression approaches, parameter sharing remains relatively unexplored. In this paper, we introduce **Fi**ne-grained **P**arameter **S**haring (FiPS), a unified compression framework that combines parameter sharing, tensor decomposition, and sparsity for achieving optimal compression. FiPS compresses transformers by factorizing MLPs concatenated across layers into a shared low-rank basis with sparse, layer-specific projection matrices. Both components are initialized by singular-value decomposition (SVD) and jointly optimized with block-wise reconstruction error minimization. As a result, FiPS enables compression of a variety of Vision Transformers (ViTs) and Large Language Models (LLMs) by 20–50% with negligible degradation in quality. Finally, we combine FiPS with Quantization Aware Training (QAT) to obtain state-of-the-art compression results on GEMMA-2 models. These results establish fine-grained parameter sharing as a practical route to compact, high-performance transformer models.

## 1 INTRODUCTION

Over the past decade, large neural networks have delivered impressive performance by scaling datasets and model sizes. However, this trend has introduced substantial computational, memory, and storage burdens, highlighting the need for efficient model compression to reduce overhead and enable deployment on resource-constrained devices such as mobile phones and embedded systems. In response, researchers have explored various strategies, including tensor decomposition, quantization, distillation, sparsity, adaptive computing methods, and *parameter sharing* (Cheng et al., 2020). While most of these techniques are well-studied and adopted, parameter sharing in neural networks remains under-explored, and has so far not been leveraged successfully to compress Transformer models despite its promise for significant parameter-count reduction.

Sharing parameters across multiple neural network layers can theoretically reduce memory usage and improve cache efficiency, thereby accelerating execution. Building on this idea, several previous works have investigated reusing entire Transformer blocks within the network architecture (Lan et al., 2020; Takase & Kiyono, 2023; Lin et al., 2023), yielding more efficient models. Although directly sharing unmodified weights across layers is promising, we hypothesize that a more fine-grained approach could lead to better compression.

Consequently, our focus shifts to sharing neurons across layers by introducing a shared basis, with each neuron expressed as a linear combination of this basis and a projection matrix. We further find that enforcing sparsity in the projection matrix is essential for the effectiveness of this approach. This insight leads to our novel parameter sharing algorithm, FiPS, which we demonstrate effectively compresses large Vision Transformers (ViTs) and Large Language Models (LLMs)[1]. Our contributions include:

- **Systematic Analysis.** We systematically explore strategies for sharing bases and neurons across transformer layers, focusing on sharing granularity and concatenation schemes. We show when neuron sharing is most effective and how global versus local and structured versus unstructured sparsity patterns contribute to further compression.

- **FiPS Algorithm[2].** Shared low-rank bases and sparse refinement factors are initialized with Singular Value Decomposition (SVD) and jointly optimized using sparse training and block-wise reconstruction-error minimization, followed by optional end-to-end fine-tuning (FT).

- **State-of-the-art ViT & LLM Compression.** FiPS delivers substantial compression with minimal performance loss, surpassing recent baselines. It shrinks DEIT-B and SWIN-L by 20–50% with <1% top-1 accuracy drop across five vision benchmarks, and compresses LLAMA-7B and LLAMA-3.1-8B by up to 40% while maintaining competitive quality on 10 NLP benchmarks.

- **Quantization-Aware Training (QAT).** We show that 3-bit QAT with FiPS compresses GEMMA-2-2B effectively, achieving a compression ratio comparable to 2-bit quantization but with markedly better language modeling, demonstrating the orthogonality of FiPS to QAT.

## 2 PARAMETER SHARING THROUGH SPARSE TENSOR DECOMPOSITION

Consider a weight matrix, $\mathbf{W} \in \mathbb{R}^{d \times p}$, which projects feature vectors from a $d$-dimensional space to a $p$-dimensional space, with neurons represented by the columns of $\mathbf{W}$. Our objective is to share weights among a subset of these $p$ neurons, reducing the number of unique neurons to $r < p$. In other words, only $r$ columns of $\mathbf{W}$ will contain unique values. These $r$ unique neurons are represented using a lookup table (basis matrix) $\mathbf{U} \in \mathbb{R}^{d \times r}$. The original matrix $\mathbf{W}$ is then reconstructed by mapping each of its $p$ columns to an $r$-dimensional one-hot vector via a projection matrix $\mathbf{V} \in \mathbb{R}^{r \times p}$, i.e., $\mathbf{W} = \mathbf{UV}$. This "one-hot" approach is illustrated in the upper part of Figure 1. However, limiting the number of unique neurons to $r$ constrains the representational capacity of $\mathbf{W}$. To address this limitation, we can increase the number of non-zero elements in $\mathbf{V}$, effectively creating combinations of the basis neurons and generating a significantly larger set of unique neuron representations, as shown in the lower part of Figure 1.

This approach can be readily extended from sharing neurons within a single weight matrix, $\mathbf{W}$, to multiple weight matrices $\mathcal{W} = \{\mathbf{W}_1, ..., \mathbf{W}_N\}$, by means of concatenation (see Figure 5). Specifically, fine-grained parameter sharing across multiple layers can be achieved by expanding the size of the projection matrix $\mathbf{V}$ and the shared basis $\mathbf{U}$. Sharing neurons between layers in this manner may be viewed as a low-rank decomposition of the matrices $\mathcal{W}$, where the first factor $\mathbf{U}$ is shared across $N$ layers and the second, layer-specific factor $\mathbf{V}$ is sparse. Therefore, existing low-rank decomposition techniques can be employed to obtain an optimal shared orthogonal basis, and sparsity in the projection matrices can be induced using current pruning and sparse training methods.

In the following sections, we investigate optimal layer-tying strategies within our framework, using a 12-block DEIT-B encoder with a single MLP module in each block, pretrained on ImageNet-1k (Deng et al., 2009). We focus on MLP modules accounting for the majority of parameters (e.g., 70.5% in GEMMA-2-9B (Team et al., 2024)) and composed of two fully connected (FC) layers (i.e., FC-1 and FC-2 for DEIT-B) of dimensions $\mathbb{R}^{d \times p}$ and $\mathbb{R}^{p \times d}$ with $p = 4d$. The "parameter budget" denotes the fraction of nonzero parameters retained after truncated SVD and sparsification, i.e., 25% keeps one-quarter of each MLP's weights. We measure the overall compression ratio as the percentage reduction in model size in bits, including sparsity metadata, i.e., a 20% compression reduces storage by 20%.

### 2.1 OPTIMAL SPARSITY FOR TENSOR DECOMPOSITION

Before implementing parameter sharing through shared bases, we decompose individual FC layers of MLP modules using a truncated SVD at 25% parameter budget. Subsequently, sparsity is introduced

---

[1]All model links are available in Appendix A.2.

[2]Source code will be released upon acceptance.

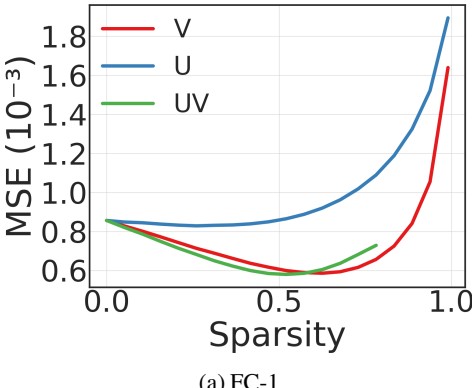 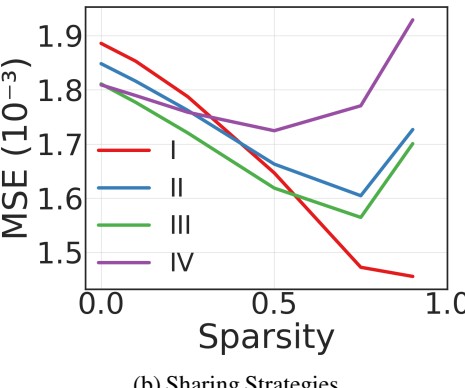

(a) FC-1            (b) Sharing Strategies

Figure 2: **Initial Experiments.** (a) Reconstruction error with varying levels of sparsity on different factors of the low-rank decomposition of FC-1 under 25% parameter budget. Results are analogous for FC-2, i.e., inducing sparsity on the larger factor yields a higher rank and, thus, lower reconstruction error. (b) Mean reconstruction error across four FCs of two distinct decoder blocks' MLPs under various parameter sharing schemes and sparsities. See § 2.2 for details.

by pruning low-magnitude values. Specifically, we examine sparsity induction in: (1) $\mathbf{U}$, (2) $\mathbf{V}$, and (3) both $\mathbf{U}$ and $\mathbf{V}$. Throughout this process, we vary the sparsity levels of the matrices while maintaining a constant total number of non-zero parameters. The resulting reconstruction errors are presented in Figure 2a. Our experiments show that the lowest reconstruction errors occur at sparsity levels between 60% and 80%, as confirmed by a sparsity sweep on ImageNet-1k with DEIT-B (see Appendix A.6), especially when sparsity is imposed on the larger factor matrix $\mathbf{V}$. We attribute this effectiveness to the higher redundancy in larger matrices, facilitating more efficient pruning.

## 2.2   WEIGHT CONCATENATION AND SHARING DIMENSIONS

We investigate parameter sharing across multiple layers by analyzing four fully connected (FC) layers drawn from two distinct MLP modules. To align their dimensionalities, we transpose the second FC layer of each module, representing every layer as $\mathbf{W} \in \mathbb{R}^{d \times 4d}$. We then examine four concatenation strategies for constructing a shared weight block $\mathbf{W}_s$:

    (I) **Full long-axis concatenation**: concatenate all four $\mathbf{W}$ along their longer dimension, yielding $\mathbf{W}_s \in \mathbb{R}^{d \times 16d}$.

    (II) **Module-wise long + inter-module short**: within each MLP, concatenate its two FC layers along the long axis, then concatenate the resulting blocks across modules along the short axis, producing $\mathbf{W}_s \in \mathbb{R}^{2d \times 8d}$.

    (III) **Module-wise short + inter-module long**: within each MLP, concatenate its two FC layers along the short axis, then concatenate the resulting blocks across modules along the long axis, again yielding $\mathbf{W}_s \in \mathbb{R}^{2d \times 8d}$.

    (IV) **Full short-axis concatenation**: concatenate all four layers along their shorter dimension to form $\mathbf{W}_s \in \mathbb{R}^{4d \times 4d}$.

For each concatenated block $\mathbf{W}_s$, we perform truncated SVD to retain the top $r$ singular vectors, followed by sparsification of the right singular matrix $\mathbf{V}$ via magnitude pruning of its largest entries (see § 2.1). Reconstruction is then obtained using the resulting shared basis, and mean squared error (MSE) is reported in Figure 2b. Empirically, concatenation along the longer dimension consistently achieves the lowest reconstruction error—particularly under high sparsity—thereby motivating our choice of full long-axis concatenation throughout. Further implementation details on the concatenation schemes are provided in Appendix A.1.1.

## 2.3   PARAMETER SHARING ACROSS LAYERS

This section examines redundancy and interdependencies among MLP modules to identify optimal parameter sharing groupings. We first decompose each module individually at rank $r = 180$ and plot the resulting mean squared error in the lower panel of Figure 3a. The error increases almost

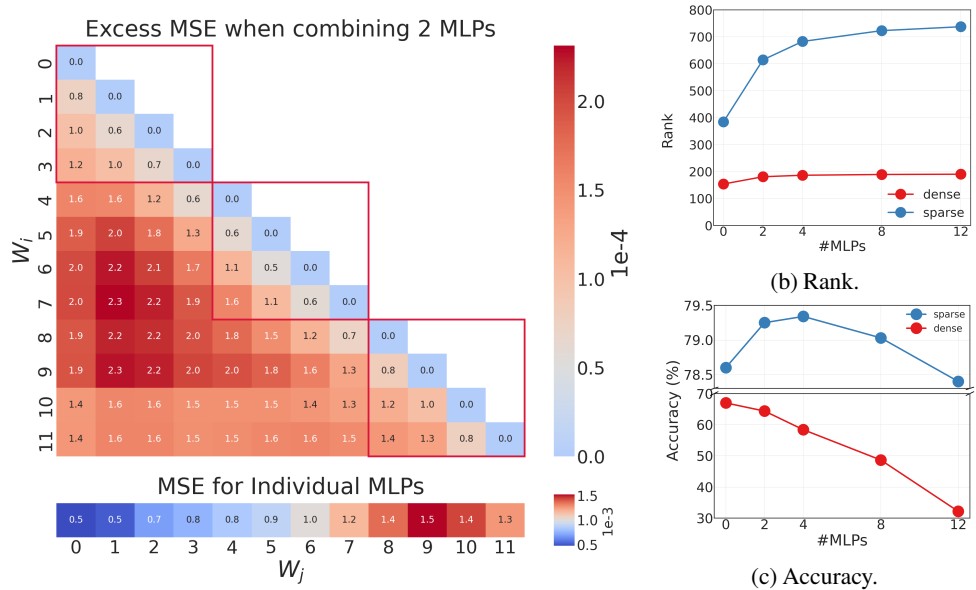

(a) MSE when compressing MLP pairs of different blocks.

(b) Rank.

(c) Accuracy.

Figure 3: **Parameter Sharing Groups.** (a top) Mean squared error (MSE) increases when sharing **U** across different MLP modules, with red squares indicating that sharing adjacent modules enhances reconstruction. (a bottom) MSE for compressing individual MLP modules, demonstrating that sharing **U** among consecutive layers typically results in the lowest error. (b) For a fixed parameter budget, the rank of the shared basis **U** stabilizes around four MLP modules, aligning with the optimal group size (c) for maximizing accuracy in the DEIT-B model.

monotonically with module depth, indicating that deeper layers require greater representational capacity. Next, we evaluate pairwise parameter sharing between modules $i$ and $j$ through a shared basis **U**. Parameter sharing reduces the total number of unique parameters, but increases the reconstruction error for each module. We denote the average error due to parameter sharing between modules $i$ and $j$, $MSE_{i,j}^{\downarrow}$, as the average increase in compression error:

$$MSE_{i,j}^{\downarrow} = \frac{(MSE_{i,j} - MSE_i) + (MSE_{j,i} - MSE_j)}{2},$$

where $MSE_{i,j}$ indicates the error of module i when sharing a basis with module j. Figure 3a shows that adjacent modules exhibit the smallest error due to parameter sharing, motivating the practice of grouping consecutive layers for parameter sharing.

We, then, explore the effect of grouping multiple MLP modules. Increasing the size of the group allows a higher rank for the shared basis **U**, as shown in Figure 3b. This benefit is most pronounced when the projection matrices **V** are sparsified. However, a higher rank does not always improve task performance, since **U** must capture a larger set of neurons. Figure 3c demonstrates that sharing across four consecutive MLP modules yields the highest post-compression accuracy.

Overall, our results reveal that (i) deeper modules require greater capacity when compressed in isolation—an effect we confirm with global pruning experiments detailed later; (ii) parameter sharing between adjacent layers curbs the rise in reconstruction error; and (iii) for DEIT-B, setting the grouping hyper-parameter to four consecutive MLP modules ($\beta = [4,4,4]$) achieves the best trade-off between basis rank and sparsity in FiPS.

## 3 FINE-GRAINED PARAMETER SHARING

The insights from § 2 motivate FiPS, an efficient parameter-sharing algorithm grounded in sparse tensor decomposition. FiPS consists of three stages:

1. **Shared Initialization.** Tie the FC layers within each MLP group and apply truncated SVD to their concatenation (see Figure 5), producing a shared basis **U** and projection matrices $\{\mathbf{V}_i\}$.

2. **Local Error Minimization.** With a small calibration dataset $D$ (§ 4.1), optimize $\mathbf{U}$ and each $\mathbf{V}_i$ to minimize the $\ell_2$ discrepancy between original and compressed activations, while enforcing target sparsity in $\mathbf{V}_i$.

3. **Global Error Minimization (Optional).** Fine-tune the compressed model end-to-end under a dynamic sparse training regime to recover performance lost at higher compression ratios.

**Shared Initialization.** We begin by compressing the pre-trained model through parameter sharing, achieved by concatenating and decomposing multiple FC layers simultaneously. For higher parameter budgets and sparsity levels (e.g., 26.5% and 75%, respectively, for DEIT-B), the rank of the shared factor $\mathbf{U}$ can exceed the model dimension $d$. In such cases, we grow the matrices $\mathbf{U}$ and $\mathbf{V}$ similar to the approach in Net2Net (Chen et al., 2016). However, unlike Net2Net, which makes a copy of each neuron and halving its float value, we grow $\mathbf{U}$ by appending zeros rather than splitting each neuron. For $\mathbf{V}$, we select the top-$k$ neurons with the highest singular values (i.e., $k = r - d$) and multiply them by $1/\tau$, where $\tau$ is treated as a hyper-parameter (see Appendix A.1.2).

Formally, the parameters of a group of FC layers across MLP modules, $\mathbf{W}_1, \mathbf{W}_2, ..., \mathbf{W}_N$, are concatenated into $\mathbf{W_s} = [\mathbf{W}_1; \mathbf{W}_2; ...; \mathbf{W}_N]$, where $\mathbf{W}_i \in \mathbb{R}^{d \times p}$ [3]. We then apply truncated SVD, $\mathbf{W_s} = \mathbf{U} \mathbf{\Sigma} \hat{\mathbf{V}}$, to obtain a low-rank approximation of the parameters, where $\mathbf{U} \in \mathbb{R}^{d \times r}$, $\mathbf{\Sigma} \in \mathbb{R}^{r \times r}$, and $\hat{\mathbf{V}} \in \mathbb{R}^{r \times (N \cdot p)}$. The factor $\mathbf{U}$ is shared among all layers within the group and remains dense due to its small size. Next, we multiply $\hat{\mathbf{V}}$ by the singular values to obtain the projection matrix $\mathbf{V} = \mathbf{\Sigma} \hat{\mathbf{V}}$. Finally, the weights are reconstructed as $\mathbf{W'}_i = \mathbf{U} \mathbf{V}_i$, where each $\mathbf{V}_i$ is a slice of $\mathbf{V}$ corresponding to the weight matrix $\mathbf{W}_i$.

**Local Error Minimization** For the second phase of FiPS, we compute the input and output activations of the original FC layers using a calibration dataset $D$, described in § 4.1. We use these activations to optimize the compressed layers and minimize the $\ell_2$-*loss* between the original and compressed layers' activations:

$$\underset{\mathbf{U}, \mathbf{V_i}, ..., \mathbf{V_N}}{\mathrm{argmin}} \sum_{i}^{N} \| \mathbf{W}_i \mathbf{X}_i - \mathbf{U} \mathbf{V}_i \mathbf{X}_i \|_2^2, \tag{1}$$

where $\mathbf{X}_i$ is the inputs to the $i^{\text{th}}$ original FC layer.

We explore several sparse training and pruning strategies to identify a sparse $\mathbf{V}$ during optimization: (a) *Static Sparsity*, which fixes the sparsity pattern by retaining the top-magnitude connections before training (Hoefler et al., 2021); (b) *Gradual Magnitude Pruning (GMP)* (Zhu & Gupta, 2017), which progressively increases sparsity by updating the mask every $T$ steps according to the cubic schedule of Kurtic et al. (2023); and (c) *RigL* (Evci et al., 2021), which initializes as in (a) but updates the sparse connectivity every $\Delta T$ steps using both gradient and magnitude information. We adopt *GMP* as our final strategy due to its superior performance, achieving up to 4% higher top-1 accuracy on ImageNet-1k compared to the closest sparsity baseline. A detailed comparison of sparsification methods is provided in Table 6.

During this stage, the parameters are shared across multiple MLP modules, gradients can be computed one MLP module at a time. Therefore, optimization requires significantly fewer resources compared to end-to-end fine-tuning.

**Global Error Minimization.** In this optional stage, we fine-tune the learned parameter sharing scheme end-to-end to further improve performance and leverage the masks learnt via GMP. Because the factors $\mathbf{V}_i$ are sparse, we employ the dynamic sparse training method, *RigL*, during this stage as it performs slightly better than *Static Sparsity* as discussed in § 4.1.

## 4 MAIN RESULTS

### 4.1 VISION TRANSFORMERS

**Experimental Setup.** We evaluate FiPS on DEIT-B (Touvron et al., 2021) and SWIN-L (Liu et al., 2021). Each model is calibrated on 2,560 ImageNet-1k images for 20 epochs, which is sufficient for convergence; calibration completes in under one hour on an NVIDIA A6000. For parameter sharing, we group every four MLP modules in DEIT-B. In SWIN-L, which consists of four stages with 2, 2, 18, and 2 encoder blocks, we share across entire stages for the three small ones and use groups

---

[3]The 2$^{\text{nd}}$ FC is transposed to match the dimensions of the 1$^{\text{st}}$.

---

**Algorithm 1 Fine-grained Parameter Sharing**

---

**Require:** MLP parameters $\mathbf{W}_1, \cdots, \mathbf{W}_N \in \mathbb{R}^{d \times p}$, MLP inputs $\mathbf{A}_i$ and MLP function $\mathbf{f}(\mathbf{W}_i, \mathbf{A}_i)$,
    Target rank $r$, Learning Rate $\eta$, Steps $\mathbf{T}$.
1: $\mathbf{U}, [\mathbf{V}_1, \mathbf{V}_2, \cdots, \mathbf{V}_N] \leftarrow TruncatedSVD([\mathbf{W}_1; \mathbf{W}_2; \cdots; \mathbf{W}_N], k = r)$
2: **for** each training iteration $t = 1$ to $T$ **do**
3:      $\mathbf{G}_\mathbf{U} = 0$                                             ▷ Gradient accumulator for $\mathbf{U}$
4:      **for** each block $i$ **do**
5:          $\mathbf{V}_i \leftarrow Sparsify(\mathbf{V}_i, t)$                        ▷ Potentially increase or adjust sparsity
6:          $L_i \leftarrow MSE\_loss(\mathbf{f}(\mathbf{W}_i, \mathbf{A}_i), \quad \mathbf{f}(\mathbf{U}\mathbf{V}_i, \mathbf{A}_i))$
7:          $\mathbf{V}_i \leftarrow \mathbf{V}_i - \eta \nabla_{\mathbf{V}_i} L_i$
8:          $\mathbf{G}_\mathbf{U} \leftarrow \mathbf{G}_\mathbf{U} + \nabla_\mathbf{U} L_i$
9:      **end for**
10:     $\mathbf{U} \leftarrow \mathbf{U} - \dfrac{\eta}{N} \mathbf{G}_\mathbf{U}$
11: **end for**
12: **return** $\mathbf{U}, [\mathbf{V}_1, ..., \mathbf{V}_N]$

---

Table 1: **ViT Compression Results.** ImageNet-1k Top-1 validation accuracy of DEIT-B (81.85%) (Touvron et al., 2021) and SWIN-L (86.24%) (Liu et al., 2021) compressed using FiPS and AAFM/GFM across parameter budgets. The results compare layer-wise (FiPS) and global error minimization (FiPS + FT). AAFM/GFM[†] results are from Yu & Wu (2023).

| Parameter Budget | 10% | | 25% | | 40% | | 50% | | 75% | |
|---|---|---|---|---|---|---|---|---|---|---|
| Method / Model | DEIT | SWIN | DEIT | SWIN | DEIT | SWIN | DEIT | SWIN | DEIT | SWIN |
| AAFM [†] | – | – | – | – | 80.33 | – | 81.21 | 85.04 | 81.76 | 85.94 |
| GFM [†] | – | – | – | – | 81.28 | – | 81.62 | 85.44 | 81.83 | 86.01 |
| FiPS (ours) | 70.04 | 74.04 | 80.64 | 84.78 | **81.69** | **85.69** | 81.83 | 85.99 | 81.82 | 86.21 |
| FiPS + FT (ours) | **77.26** | **82.13** | **81.31** | **85.16** | 81.54 | 85.68 | 81.54 | **85.99** | 81.82 | **86.22** |

of six blocks in the larger stage. Additional hyper-parameters and sensitivity analyses are provided in Appendices A.6 and A.7.

**ImageNet-1k.** We compare FiPS to Adaptive Atomic Feature Mimicking (AAFM), which compresses output activations rather than weights, and Global Feature Mimicking (GFM), which fine-tunes the compressed network (Yu & Wu, 2023). Both FiPS and FiPS+FT match AAFM and GFM in compute and memory budgets. At a 40% parameter budget, FiPS outperforms AAFM by 1.36 points and GFM by 0.41 points despite GFM's higher cost (see Table 1). This holds across all budgets: FiPS consistently achieves the highest accuracy with the lowest overhead. Notably, a 10% parameter budget corresponds to roughly a 50% compression ratio, where FiPS+FT yields particularly strong gains.

**Transfer Learning** For transfer learning, we fine-tune for 100 epochs on CIFAR-100, Flowers102, Oxford-III-Pets, and iNaturalist 2019 (Krizhevsky, 2009; Nilsback & Zisserman, 2008; Parkhi et al., 2012; Van Horn et al., 2018), following Yu & Wu (2023). We use *AdamW* (Loshchilov & Hutter, 2019) with learning rates chosen from 12 log-spaced values. The models compressed with FiPS transferred significantly better, as shown in Table 2a.

**Latency and Memory Profiling** Structured sparsity patterns enable efficient hardware implementations with minimal quality impact, as demonstrated in Table 2b using *NMGMP*. With 2:4 structured FiPS, the degradation remains just above 1% at 10% and 25% MLP parameter budgets; for all other settings, the impact is negligible. We further evaluate FiPS with alternative structured sparsity pruners—*STE*, *SR-STE* (Zhou et al., 2021), and *NMSRigL* (Lee et al., 2023; Lasby et al., 2024)—in Table 7. Latency and memory profiling with the optimal *NMGMP*+FiPS setup leverages NVIDIA's tensor core support for 2:4 sparsity (Mishra et al., 2021) on GPUs and Neural Magic's DeepSparse Engine (Neural Magic, 2021) on CPUs, as shown in Figure 7 and detailed in Appendix A.8.

Table 2: (a) Comparison of Top-1 accuracy between the original DEIT-B and its compressed counterparts using GFM and FiPS across different parameter budgets. Results for GFM[†] and Original[†] are sourced from Yu & Wu (2023) and Touvron et al. (2021). (b) ImageNet-1k Top-1 accuracy (%) of DEIT-B (81.85%) (Touvron et al., 2021) using FiPS with 2:4 structured sparsity (via N:M Structured GMP (Lee et al., 2023)) compared to unstructured sparsity, both at 50% sparsity.

| | (a) | | | | | | | (b) | | | | |
|---|---|---|---|---|---|---|---|---|---|---|---|---|
| | Original[†] | GFM[†] | | FiPS+RigL FT (ours) | | | | P. Budget | 10% | 25% | 40% | 50% | 75% |
| P. Budget | 100 | 40% | 50% | 25% | 40% | 50% | | | | | | |
| | | | | | | | | 2:4 FiPS | 52.36 | 76.88 | 80.59 | 81.31 | 81.51 |
| | | | | | | | | FiPS | 54.00 | 77.56 | 80.94 | 81.63 | 81.77 |
| CIFAR-100 | 90.99 | 90.17 | 90.67 | 90.88 | 91.24 | **91.33** | | | | | | |
| Pets | **94.74** | 93.95 | 94.22 | 94.19 | 94.52 | 94.41 | | | | | | |
| Flowers102 | 97.77 | 97.02 | 97.45 | 97.84 | 98.14 | **98.37** | | | | | | |
| iNaturalist 2019 | 77.39 | 77.13 | 77.56 | 77.26 | 77.58 | **77.69** | | | | | | |

These results demonstrate that FiPS effectively compresses models while enhancing both memory efficiency and computational speed. For instance, with a batch size of 64, employing FiPS with 2:4 sparsity at a 22.14% parameter budget results in a $1.31\times$ speed-up on the NVIDIA A4000 and reduces maximum VRAM allocation to approximately $0.79\times$ of the original requirement during inference.

## 4.2 LARGE LANGUAGE MODELS

**Experimental Setup.** We evaluate FiPS on three publicly available, pre-trained LLMs: LLAMA-7B (Touvron et al., 2023), LLAMA-3.1-8B (Grattafiori et al., 2024), and the instruction-tuned GEMMA-2-2B (Team et al., 2024). GEMMA-2-2B is included solely to demonstrate the orthogonality of FiPS with QAT, while the other models are compared against the baselines described below. Unlike ViTs, these LLMs feature three fully connected layers per MLP in their decoder blocks. For parameter sharing, all FC layers from MLPs within the same group are concatenated. The number of MLPs per group is again treated as a hyper-parameter and detailed in appendix A.1.1. Calibration and optimization follow the ViT protocol (as described in § 4.1): activations are collected from $8,192 \times 20$ SlimPajamas (Soboleva et al., 2023) tokens, and block-wise error minimization is performed for 40 epochs on a single NVIDIA A100 (80GB), completing within 10 hours per model.

**Baselines.** We benchmark FiPS against three recent SVD-based LLM compression methods. Activation-aware SVD (ASVD) scales each weight matrix by activation statistics to mitigate outliers and reduce layer sensitivity prior to SVD (Yuan et al., 2023). SVD-LLM employs truncation-aware whitening and, optionally, sequential low-rank updates to minimize truncation error (Wang et al., 2024). Its successor, SVD-LLM V2, further enhances performance through layer-wise and loss-aware rank allocation (Wang et al., 2025). All baseline metrics are taken directly from the original publications. Importantly, we do not apply any fine-tuning or Low-Rank Adaptation (LoRA) (Hu et al., 2021); accordingly, our comparisons with SVD-LLM and SVD-LLM V2 exclude LoRA enhancements.

**Evaluation.** In addition to reporting perplexity on WikiText-2 (Merity et al., 2016) and C4 (Dodge et al., 2021), we evaluate six classification benchmarks—OpenbookQA (Mihaylov et al., 2018), ARC-easy (Clark et al., 2018), Winogrande (Sakaguchi et al., 2021), Hellaswag (Zellers et al., 2019), PIQA (Bisk et al., 2020), and MathQA (Amini et al., 2019)—as well as two generation tasks, TruthfulQA (Lin et al., 2022) and GSM8K (Cobbe et al., 2021), using the LM-Evaluation-Harness (Sutawika et al., 2023). Table § 4.2 shows that, at a 20% compression ratio, FiPS attains the lowest perplexity (PPL↓) on both WikiText-2 and C4 for LLAMA-7B and LLAMA-3.1-8B, while matching or surpassing the best classification and generation scores across all downstream tasks—outperforming all SVD-based baselines. We further extend the C4 perplexity results to a 40% compression ratio in Table 5.

**Quantization-Aware Training (QAT).** To assess FiPS under low-precision regimes, we apply QAT to the MLP layers of GEMMA-2-2B for a fair comparison. Table § 4.2 reports WikiText-2 perplexity for 4- and 2-bit QAT—corresponding to $4\times$ and $8\times$ MLP-layer compression relative to a `bfloat16` baseline—as well as FiPS combined with 3-bit QAT. While 4-bit QAT matches `bfloat16` performance, 2-bit QAT suffers severe degradation (PPL = 41.86; $\sim 270\%$ increase).

| | Method | WikiText-2↓ | C4↓ | Openb. | ARC-e | WinoG. | HellaS. | PIQA | MathQA | **Avg.↑** | TruthfulQA↑ | GSM8K↑ |
|---|---|---|---|---|---|---|---|---|---|---|---|---|
| LLAMA-7B | Original | 5.68 | 7.34 | 0.34 | 0.75 | 0.70 | 0.57 | 0.79 | 0.27 | 0.57 | 0.30 | 0.09 |
| | ASVD | 11.14 | 15.93 | 0.29 | 0.53 | 0.64 | 0.41 | 0.68 | 0.17 | 0.45 | 0.21 | 0.04 |
| | SVD-LLM | 7.94 | 15.84 | 0.31 | 0.71 | 0.68 | 0.49 | 0.71 | 0.22 | 0.52 | 0.24 | 0.06 |
| | SVD-LLM V2 | 7.12 | 10.47 | **0.32** | **0.72** | **0.70** | 0.52 | 0.75 | 0.24 | 0.54 | **0.27** | **0.07** |
| | FiPS (ours) | **6.06** | **8.10** | 0.32 | 0.72 | 0.70 | **0.56** | 0.78 | 0.26 | 0.56 | 0.27 | 0.07 |
| LLAMA-3.1-8B | Original | 6.14 | 9.47 | 0.35 | 0.80 | 0.73 | 0.60 | 0.80 | 0.40 | 0.61 | 0.49 | 0.45 |
| | ASVD | 17.55 | 28.41 | 0.20 | 0.59 | 0.61 | 0.41 | 0.69 | 0.30 | 0.47 | 0.37 | 0.28 |
| | SVD-LLM | 11.82 | 20.05 | 0.29 | 0.77 | 0.64 | 0.51 | 0.72 | 0.30 | 0.54 | 0.45 | 0.31 |
| | SVD-LLM V2 | 8.01 | 11.72 | **0.33** | **0.79** | 0.70 | 0.58 | 0.77 | 0.36 | 0.59 | **0.46** | 0.40 |
| | FiPS (ours) | **6.88** | **10.78** | 0.33 | 0.79 | **0.72** | **0.59** | 0.78 | **0.38** | **0.60** | 0.46 | **0.42** |

Table 3: **LLM Compression Results.** We evaluate LLAMA-7B and LLAMA-3.1-8B at 20% compression ratio, reporting perplexity (PPL↓) on WikiText-2 and C4, macro-averaged accuracy (Avg.↑) on six classification tasks, and generation quality on TruthfulQA (BLEU↑) and GSM8K (exact match↑).

| Variant | Precision | Compression | PPL ↓ |
|---|---|---|---|
| Baseline | BF16 | 1.0× | 15.61 |
| QAT | INT4 | 4.0× | 16.86 |
| QAT | INT2 | 8.0× | 41.86 |
| FiPS | BF16 | 1.5× | 32.01 |
| FiPS+QAT | INT3 | 8.0× | 35.43 |

Table 4: **Quantization-Aware Training (QAT).** A FiPS-compressed model at $1.5\times$ MLP compression (66.67% of its original size), combined with 3-bit QAT, achieves substantially better language modeling performance while matching the effective compression of 2-bit QAT.

By contrast, augmenting 3-bit QAT with FiPS at $1.5\times$ MLP compression yields a perplexity of 35.43—substantially outperforming 2-bit QAT alone while achieving the same effective compression.

## 5 ABLATIONS

In the following, we examine the importance of various components of the FiPS algorithm when compressing DEIT-B at 25% parameter budget. First, we ablate the key components of our algorithm:

1. **Random Initialization (RI):** Using RI instead of SVD initialization results in a 1% point drop in accuracy.

2. **Global Pruning (GP):** Using GP when sparsifying our sparse factors $\mathbf{V}$ results in 0.4% point improvement over local pruning (LP), which enforces the same sparsity level for each group.

3. **Scaling Vectors (SV):** Following Liu et al. (2024), FC weights are normalized, and the magnitudes initialize the SV for neuron scaling, which enhances local pruning but is less effective than global pruning.

Moreover, we perform a sensitivity analysis using different sparsification methods, sparsity levels, calibration dataset sizes, and training durations, using the DEIT-B checkpoint trained on ImageNet-1k. GMP consistently outperforms alternatives and is adopted in our final setup. We find that 75% sparsity delivers optimal performance across calibration settings, and best with 20 epochs over 20 batches. See Appendix A.6 for full results.

**Sparsity Distribution and MSE-loss**    Initial experiments in Figure 3a reveal that later layers exhibit higher reconstruction error under uniform compression budgets, suggesting these layers benefit from a greater parameter allocation. FiPS addresses this by applying global magnitude pruning to its sparse factors, automatically directing parameters where they are most needed. As shown in Figure 4b, FiPS indeed assigns more parameters to the later layers. Furthermore, Figure 4c demonstrates a strong negative correlation ($-0.922$) between the final sparsity pattern and the MSE losses observed in Figure 3a, confirming the effectiveness of this adaptive allocation.

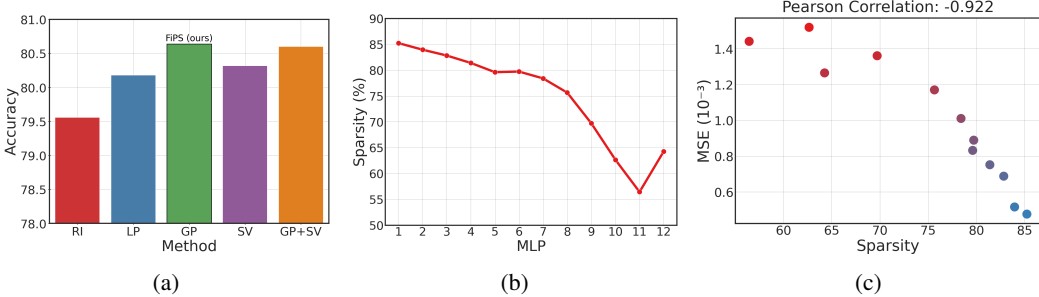

(a)    (b)    (c)

Figure 4: **DEIT-B Ablation and Global Sparsity Analysis.** (a) Component analysis of the FiPS algorithm: Random Initialization (RI), Local Pruning (LP), Global Pruning (GP), and Scaling Vectors (SV). (b) End-of-training sparsity allocation, later layers require more parameters. (c) Strong correlation between the MSE reported in Figure 3a and the parameter distribution captured by FiPS.

## 6    RELATED WORK

**Vision Transformers (ViT) & Large Language Models (LLM).**    Recent transformer architectures extend beyond the foundational ViT models (Dosovitskiy et al., 2021), which treat image patches as token sequences. DEIT (Touvron et al., 2021) enhances data efficiency with distillation tokens, while SWIN (Liu et al., 2021) introduces a hierarchical design using shifted windows. Both vision models employ two FC layers per MLP module. In contrast, decoder-only LLMs such as LLAMA (Touvron et al., 2023), LLAMA-3 (Grattafiori et al., 2024), and GEMMA-2 (Team et al., 2024) attain state-of-the-art zero-shot and instruction-following performance by using three FC layers per MLP module in each block.

**Sparsity in Neural Networks.**    Early methods involved heuristic pruning, such as removing the smallest magnitude parameters (Thimm & Fiesler, 1995). Later approaches, like GMP (Zhu & Gupta, 2017), increased the amount of pruning, while dynamic pruning with accelerated schedulers was explored by Kurtic et al. (2023). Moreover, static sparsity uses a pre-initialized mask throughout training (Hoefler et al., 2021), whereas dynamic methods, like RigL (Evci et al., 2021), adjust the sparsity pattern during training based on gradient information.

**Tensor Decomposition.**    Yu & Wu (2023) apply Adaptive Atomic Feature Mimicking (AAFM) and its global variant Global Feature Mimicking (GFM) apply truncated PCA on ViT activations, followed by fine-tuning, respectively. In LLMs, Activation-aware SVD (ASVD) (Yuan et al., 2023), truncation-aware SVD-LLM (Wang et al., 2024), and SVD-LLM V2's rank-distillation and layer-wise allocation (Wang et al., 2025) serve as our baselines.

**Parameter Sharing.**    Eban et al. (2019) introduces using a Sum-Product reducer to map shared parameters, and Obukhov et al. (2021) employs TR decomposition for shared parameters in 3D tensors. Zhang et al. (2022) proposes "Weight Multiplexing," sharing parameters between MLP modules in ViTs, alongside distillation and linear projections between transformer blocks to aid model recovery.

## 7    CONCLUSION

We presented FiPS, a unified framework for compressing transformer models via fine-grained inter-layer parameter sharing enabled by sparsity and low-rank decomposition. Our results demonstrate that FiPS achieves state-of-the-art compression–accuracy trade-offs across both ViTs and LLMs: up to 50% compression in DEIT-B and SWIN-L with under 1% top-1 accuracy loss, and up to 40% compression in LLAMA-7B and LLAMA-3.1-8B. Importantly, FiPS is fully orthogonal to quantization-aware training (QAT) and can compress GEMMA-2-2B up to 8× when combined with QAT. These findings establish parameter sharing as a powerful and competitive alternative to existing compression strategies.

While this work focuses on MLPs, our approach naturally extends to attention layers, offering an exciting direction for future research. Additional avenues include quantizing the shared bases and developing specialized kernels that keep the shared basis $U$ resident in fast memory to maximize cache efficiency. Together, these advances pave the way for even more efficient and scalable on-device inference.

ETHICS STATEMENT

This paper advances Machine Learning by introducing Fine-grained Parameter Sharing (FiPS), a model compression method that improves the efficiency of Vision Transformers (ViTs) and Large Language Models (LLMs). By leveraging parameter sharing, tensor decomposition, and sparsity, FiPS reduces computational and memory costs, enhancing AI accessibility on resource-constrained devices. While model compression promotes efficiency and sustainability, it may also enable broader AI deployment in sensitive domains with ethical concerns such as bias, misinformation, and privacy. Nevertheless, this work should not introduce new risks beyond those inherent in deep learning, but we encourage responsible deployment and ethical considerations in practice.

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

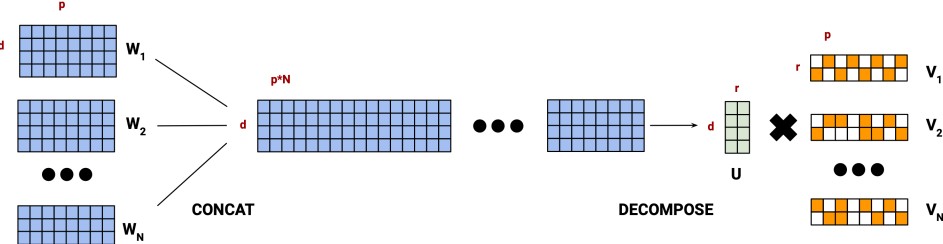

Figure 5: **Parameter Sharing Through Sparse Tensor Decomposition.** A group of FC layers are concatenated along the larger dimension, $p$, and decomposed into two matrices: a shared basis, $\mathbf{U}$, and a sparse projection matrix, which is then sliced up respectively for each layer.

## A  APPENDIX

### A.1  METHOD

In reference to § 2.2, we detail four methods for concatenating the FC weights of two MLPs (four matrices $\mathbf{W}_{ij} \in \mathbb{R}^{d \times p}$, with $p = 4d$ and $i,j \in \{1,2\}$):

I. **Full long-axis concatenation**, forming

$$\mathbf{W} = [\mathbf{W}_{11}\mathbf{W}_{12}\mathbf{W}_{21}\mathbf{W}_{22}] \in \mathbb{R}^{d \times 16d}.$$

II. **Module-wise long + inter-module short:**

$$\mathbf{W}_A = [\mathbf{W}_{11}\mathbf{W}_{12}] \in \mathbb{R}^{d \times 8d}$$

$$\mathbf{W}_B = [\mathbf{W}_{21}\mathbf{W}_{22}] \in \mathbb{R}^{d \times 8d}$$

$$\mathbf{W} = \begin{bmatrix} \mathbf{W}_A \\ \mathbf{W}_B \end{bmatrix} \in \mathbb{R}^{2d \times 8d}.$$

III. **Module-wise short + inter-module long:**

$$\mathbf{W}_C = [\mathbf{W}_{11}\mathbf{W}_{21}] \in \mathbb{R}^{2d \times 4d}$$

$$\mathbf{W}_D = [\mathbf{W}_{12}\mathbf{W}_{22}] \in \mathbb{R}^{2d \times 4d}$$

,

$$\mathbf{W} = [\mathbf{W}_C\mathbf{W}_D] \in \mathbb{R}^{2d \times 8d}.$$

IV. **Full short-axis concatenation**, yielding

$$\mathbf{W} = \begin{bmatrix} \mathbf{W}_{11} \\ \mathbf{W}_{12} \\ \mathbf{W}_{21} \\ \mathbf{W}_{22} \end{bmatrix} \in \mathbb{R}^{4d \times 4d}.$$

These configurations are evaluated in  Figure 2b and discussed in  § 2.2.

The overall method is depicted in Figure 5, and further details are explained below.

### A.1.1 MLP CONCATENATION STRATEGIES

### A.1.2 GROWING NEURONS IN SHARED BASES AND SPARSE FACTORS

As discussed in § 3, high parameter budgets and sparsity levels (e.g., 26.5% parameter budget, 75% sparsity and groups of four blocks in DeiT-B) often result in the rank $r$ exceeding the model dimension $d$. Since SVD yields only $d$ initialization directions, we investigate three methods to initialize the remaining $k = r - d$ dimensions:

1. **Random Growth:** Initialize new neurons in $\mathbf{U}$ to zero and in $\mathbf{V}$ randomly using He et al. (2015);

2. **Neuron Splitting:** Duplicate the top $k$ neurons of $\mathbf{U}$ and halve the top $k$ neurons of $\mathbf{V}$, following Chen et al. (2016);

3. **Hybrid Initialization:** Initialize new neurons in $\mathbf{U}$ to zero and derive those in $\mathbf{V}$ from the top $k$ neurons, normalized by $\tau$. This method minimizes the immediate impact of new neurons in $\mathbf{V}$, allowing their gradual reactivation, as proposed by Evci et al. (2022).

After performing a hyper-parameter sweep for $\tau$, hybrid initialization outperformed the alternatives, achieving 1% and 2% higher accuracy than methods (1) and (2), respectively.

### A.2 MODEL LINKS

- DEIT-B (Touvron et al., 2021): https://huggingface.co/facebook/deit-base-patch16-224
- SWIN-L (Liu et al., 2021): https://huggingface.co/microsoft/swin-large-patch4-window7-224
- GEMMA-2-2B-IT (Team et al., 2024): https://huggingface.co/google/gemma-2-2b-it
- GEMMA-2-9B (Team et al., 2024): https://huggingface.co/google/gemma-2-9b
- LLAMA-7B (Grattafiori et al., 2024): https://huggingface.co/huggyllama/llama-7b
- LLAMA-3.1-8B (Grattafiori et al., 2024): https://huggingface.co/meta-Llama/Llama-3.1-8B

### A.3 FURTHER LLM RESULTS

| Variant | Compression Ratio | PPL↓ |
|---|---|---|
| Original | 0% | 7.34 |
| SVD-LLM | 20% | 15.84 |
| SVD-LLM V2 | 20% | 11.72 |
| FiPS (ours) | 20% | **8.10** |
| SVD-LLM | 40% | 75.42 |
| FiPS (ours) | 40% | **10.57** |

Table 5: Perplexity on C4 for Llama7B under different compression ratios applied with FiPS and the baselines.

### A.4 DIFFERENT SPARSIFICATION METHODS

In addition to *GMP*, we evaluated *Dense* tensor decompositions (i.e., without sparsity on the $\mathbf{V}$ factors) and other sparse training techniques, specifically *Static Sparsity* and *RigL*. The results are summarized in Table 6. For DEIT-B, *RigL* consistently outperforms both *Dense* and *Static Sparsity* across parameter budgets ranging from 10% to 50%. At higher parameter budgets, all methods converge to similar

Table 6: **Sparsification Method and FiPS Generalization Performance**. ImageNet top-1 validation accuracy (%) of DEIT-B (81.85%) (Touvron et al., 2021) and SWIN-L (86.24%) (Liu et al., 2021) models compressed with FiPS using different sparsity methods: RigL (Evci et al., 2021) and static sparsity.

| Parameter Budget | 10% | | 25% | | 40% | | 50% | | 75% | |
|---|---|---|---|---|---|---|---|---|---|---|
| Method / Model | DEIT | SWIN | DEIT | SWIN | DEIT | SWIN | DEIT | SWIN | DEIT | SWIN |
| Dense | 15.35 | 3.61 | 65.71 | 60.31 | 74.33 | 80.61 | 79.22 | 83.59 | 81.36 | 85.64 |
| Static Sparsity | 65.26 | 65.6 | 80.06 | 84.37 | 81.48 | **85.69** | 81.70 | 85.98 | **81.86** | **86.23** |
| RigL | 66.67 | 70.96 | 80.31 | 84.57 | 81.50 | 85.59 | 81.65 | 85.91 | 81.82 | 86.20 |
| GMP (FiPS) | **70.04** | **74.04** | **80.64** | **84.78** | **81.69** | 85.69 | **81.83** | **85.99** | 81.82 | 86.21 |

| Compression Ratio | 10% | 25% | 40% | 50% | 75% |
|---|---|---|---|---|---|
| STE | 42.89 | 73.26 | 78.26 | 79.36 | 78.89 |
| SR-STE | 45.31 | 75.53 | 79.71 | 80.68 | 81.24 |
| NMSRigL | 44.87 | 75.71 | 79.97 | 80.99 | 81.40 |
| NMSGMP | 52.36 | 76.88 | 80.59 | 81.31 | 81.51 |
| FiPS (50% Sparsity) | 54.00 | 77.56 | 80.94 | 81.63 | 81.77 |
| FiPS (75% Sparsity) | 70.04 | 80.64 | 81.69 | 81.83 | 81.82 |

Table 7: **Structured Sparsity Performance.** ImageNet top-1 accuracy (%) of DEIT-B (81.85%) (Touvron et al., 2021) for various structured sparsification methods at 50% and 75% sparsity, compared to Unstructured FiPS. Methods include Straight Through Estimator (*STE*), Sparse-Refined STE, N:M Structured RigL (*NMSRigL*), and N:M Structured GMP (*NMSGMP*) at 50% sparsity, corresponding to 2:4 structures (Lee et al., 2023; Zhou et al., 2021; Lasby et al., 2024).

accuracies approaching the original model's performance. In the case of SWIN-L, *RigL* surpasses *Dense* and *Static Sparsity* at 10% and 25% parameter budgets. However, at higher parameter budgets, *Static Sparsity* achieves slightly higher accuracies. Detailed results on SWIN-L are presented in Table 6.

## A.5 STRUCTURED SPARSITY

We evaluate the generalization performance of FiPS using structured sparsity, with results presented in Table 2b. The methods evaluated include the Straight Through Estimator (*STE*), which employs top-$k$ weight magnitude selection, projects parameters into a sparse subspace during training, and applies gradients to dense parameters through a gradual pruning schedule; the Sparse-Refined-STE (*SR-STE*) (Zhou et al., 2021), which mitigates the adverse effects of approximated gradients; N:M Structured RigL (*NMSRigL*) and N:M Structured GMP (*NMSGMP*) (Lee et al., 2023; Lasby et al., 2024), where N:M specifies the sparsity pattern of the weight matrix (e.g., a 50% sparsity in FC matrices of size $d \times 4d$ corresponds to a 2:4 structure).

## A.6 SENSITIVITY ANALYSIS

**Calibration Dataset Size and Training Length.** We examine how the number of calibration batches and training epochs affects performance using a fixed batch size of 128. To ensure at least one example from each category, we begin with a minimum of 10 batches and also evaluate 20, 40, and 80 batches. After filtering out any configurations that are more than 0.25% below the highest accuracy, we adopt the most efficient setting of 20 epochs over 20 batches for all reported results, as shown in Figure 6b.

**Optimal Sparsity for Sparse Factors** We compressed the DEIT-B model, as described in § 4.1, using sparsity levels ranging from 50% to 96.9% as shown in Figure 6a. The best performance was observed at 75% sparsity as shown in Figure 6a. While increasing sparsity to 87% yielded similar accuracy, lowering it to 50% resulted in a notable drop in performance, likely due to a significant reduction in rank.

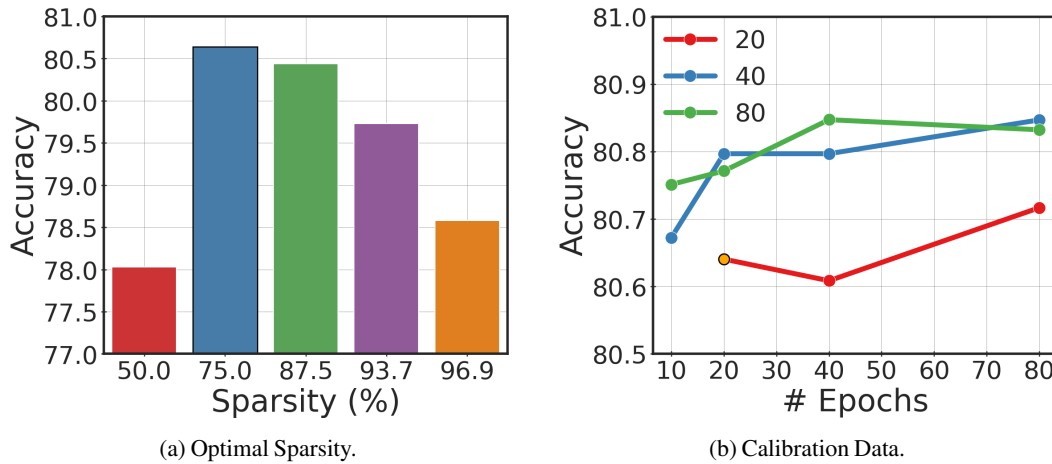

(a) Optimal Sparsity.      (b) Calibration Data.

Figure 6: **Sensitivity Analysis.** (a) Impact of sparsity levels on DEIT-B accuracy. (b) Effect of calibration data volume and training duration.

Table 8: **Block–grouping sweep for GEMMA-2-2B.** Each row lists the candidate $\beta$ and the resulting validation perplexity (PPL) on WikiText-2. Baseline (no sharing) PPL is $15.61$; lower is better.

| Config. | $\beta$ list | PPL $\downarrow$ |
|---|---|---|
| 1 | $[4,4,5,5,4,4]$ | $21.42$ |
| 2 | $[1,4,4,4,4,4,4,1]$ | $\approx 23$ |
| 3 | $[2,3,4,4,4,4,3,2]$ | $\approx 22$ |
| 4 | $[2,6,6,6,6]$ | $21.81$ |
| 5 | $[3,5,5,5,5,3]$ | $21.86$ |

## A.7 HYPER-PARAMETERS

### A.7.1 ABLATION ON THE BLOCK–GROUPING HYPER-PARAMETER $\beta$

**Definition.** We now define $\beta$ as an *ordered list* whose $i^{\text{th}}$ element gives the number of consecutive decoder blocks whose MLP weights are tied in the $i^{\text{th}}$ parameter-sharing group:

$$\beta = \big[\beta_1, \beta_2, ..., \beta_G\big], \quad \sum_{g=1}^{G} \beta_g = L,$$

where $L$ is the total number of decoder blocks. Self-attention parameters remain untied in all experiments. For every architecture we sweep over a handful of plausible $\beta$ lists and keep the one with the lowest validation perplexity (PPL) after compression.

**Block Groups of ViTs.** Using the list–valued notation for $\beta$ introduced above, we set

$$\beta_{\text{DEIT-B}} = [4,4,4],$$

i.e. three groups of four consecutive blocks (each block contains one MLP).

The depth pattern of SWIN-L is $2+2+18+2$ blocks. We tie MLP weights inside every 2-block stage and split the 18-block stage into three groups of six, which gives

$$\beta_{\text{SWIN-L}} = [2,2,6,6,6,2].$$

**Gemma-2-2B.** Table 8 shows the five $\beta$ lists evaluated for GEMMA-2-2B-IT at 20 % compression. Config. 1—$\beta = [4,4,5,5,4,4]$—yields the lowest PPL and is therefore used in the main paper.

**Llama-2-7B.** With $L = 32$ decoder blocks we compared three $\beta$ lists:

| Candidate $\beta$ | Block groups (sizes) | Observation |
|---|---|---|
| $[4,4,4,4,4,4,4,4]$ | 8 groups × 4 blocks | Highest PPL |
| **$[4,6,6,6,6,4]$** | **6 groups with sizes** $(4,6,6,6,6,4)$ | **Best PPL; used in §4.1** |
| $[8,8,8,8]$ | 4 groups × 8 blocks | Slightly worse than above |

A gently varying list—with smaller groups at the extremes and larger groups in the middle—provides the best compression/accuracy trade-off.

**Llama-3.1-8B.** The 8 B model shares the same 32-layer decoder. We reused the winning $\beta$ from the 7 B sweep,

$$\beta_{\mathrm{opt}} = [4,6,6,6,6,4],$$

because (i) it keeps the total tied-parameter ratio identical and (ii) in a spot-check it preserved PPL within +3.5 of the uncompressed baseline—better than the uniform alternatives $[8,8,8,8]$ or $[4,4,4,4,4,4,4,4]$.

**Key Insights.**

- Optimal $\beta$ often starts and ends with smaller groups, echoing the intuition that early and late layers host more specialised features.
- Extremely fine-grained sharing (e.g. many 4-block groups) hurts accuracy, while overly coarse sharing (uniform 8-block groups) gives up capacity.
- For Llama models, a tapered list such as $[4,6,6,6,6,4]$ ties roughly 22–25 % of the MLP parameters yet adds only $\sim$3–4 perplexity points.

These ablations inform all main-text compression results.

A.7.2  OPTIMIZER

**ViT Compression.** To minimize local error, we employ a logarithmic grid for hyper-parameter tuning. The learning rates for Dense, Static Sparsity, GMP, and RigL are set as follows for both DEIT-B and SWIN-L:

1. Dense: $1.25 \times 10^{-4}$,
2. Static Sparsity: $2.5 \times 10^{-4}$,
3. GMP: $1 \times 10^{-3}$,
4. RigL: $1 \times 10^{-3}$.

**ViT Transfer Learning.** We use a linear grid, as some hyper-parameters are derived from the codebase of DEIT. The optimal learning rates for FiPS are:

1. CIFAR-100: $2.5 \times 10^{-5}$;
2. Flowers102: $1 \times 10^{-4}$;
3. Oxford-III-Pets: $7.5 \times 10^{-6}$;
4. iNaturalist 2019: $1 \times 10^{-4}$.

**LLMs** Eight logarithmically spaced values were swept. The final values for FiPS are presented below:

1. GEMMA-2-2B: $4.0 \times 10^{-4}$,
2. LLAMA-7B: $1.0 \times 10^{-6}$,
3. LLAMA-3.1-8B: $1.0 \times 10^{-5}$.

**Sparsifier**

GLOBAL MASK PRUNING (GMP) GMP begins with an initial sparsity level of 25%. During the training process, the sparsity is gradually increased to 50% at the 25% training mark and ultimately reaches 75% sparsity by the end of the training. The $\Delta T$ of 50 is used for update steps.

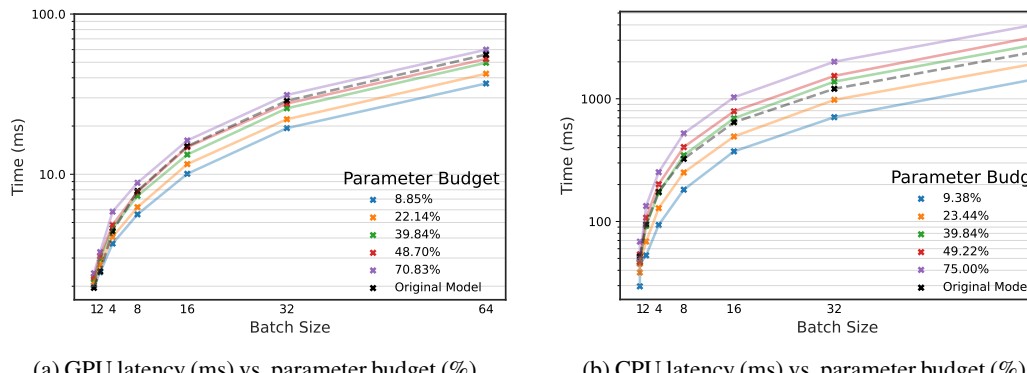

(a) GPU latency (ms) vs. parameter budget (%).  (b) CPU latency (ms) vs. parameter budget (%).

Figure 7: **DEIT-B inference latency benchmarks.** (a) End-to-end latency of 2:4 sparse FiPS on an NVIDIA A4000 for batch sizes 1–64; a 22 % parameter budget yields a 25 % speed-up once the batch size exceeds 8. (b) Latency of 75 % unstructured sparse FiPS accelerated by DeepSparse on an Intel Xeon W-2145 CPU, outperforming the dense model at every tested batch size.

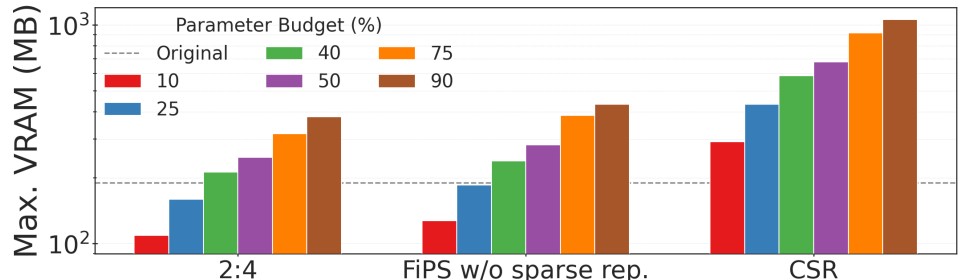

Figure 8: **DEIT-B Inference Memory Profile.** Maximum VRAM allocation at batch size 64 for 50% sparse FiPS using 2:4, strided (dense), and CSR tensor formats. At 10% and 25% parameter budgets, 2:4 sparsity reduces peak memory by 44% and 18%, respectively; CSR incurs higher overhead at modest sparsity due to index storage.

RIGL    RigL employs an initialization phase that combines pruning with a growth ratio of $0.1$ for block-wise error minimization and a growth ratio of $0.05$ for transfer learning tasks with $\Delta T$ of $50$ for growth and pruning ratio. This conservative growth ratio in transfer learning helps preserve the mask obtained during the initial training, ensuring that the learnt masks are not lost.

## A.8    LATENCY AND MEMORY PROFILING

As discussed in § 3, high levels of sparsity and parameter budgets can result in the SVD rank exceeding a model's hidden dimension. For instance, in the case of DEIT-B, achieving 75% sparsity under parameter budget constraints exceeding 26.5% and four block groups increases the rank of the shared singular vectors beyond the original model's embedding dimension. Efficient sparse operations and representations are crucial for minimizing the latency and memory overhead introduced by FiPS. Figure 7 and Figure 8 summarise the latency and memory results, respectively, for DEIT-B compressed with FiPS using 2:4 structured GMP, and highlight the resulting speed-ups and memory savings on both CPU and GPU platforms.

