# OpenReview forum: "Learning Fine-grained Parameter Sharing via Sparse Tensor Decomposition"
_ICLR.cc/2026/Conference — ICLR 2026 Conference Withdrawn Submission_

### Official Review · Reviewer_JGJ9 · 2025-10-23

**Soundness:** 2
**Presentation:** 1
**Contribution:** 1
**Rating:** 2
**Confidence:** 5

**Summary:**

This paper presents a fine-grained parameter sharing (FiPS) technique for efficient deployment of large Transformer networks. A singular value decomposition is performed on a concatenation of multiple weight matrices to identify the shared bases and coefficients, followed by a sparsification process to further reduce the number of parameters. A weight matrix is then expressed by a sparse combination of shared bases. The parameter sharing strategy is demonstrated with experimental results on vision models and language models, achieving substantial parameter reduction with minimal accuracy loss.

**Strengths:**

1. The paper provides thorough analyses of the method selections, which are clearly stated and easy to follow. In Section 2 and 3, several options for the joint SVD, pruning, and weight tying were presented, followed by intuitive and representative analyses that support the choices that were made in the paper.
2. The ImageNet-1k accuracy in Table 1 is well preserved even when the parameter budget is low. `FiPS` also improved the accuracy of LLM from the non-parameter-shared low-rank model.
3. The writing was clear and easy to follow.

**Weaknesses:**

I lean toward rejection because (1) the paper is missing an important baseline, (2) the novelty of the proposed method is limited, and (3) the experimental results are not strong enough to overcome the aforementioned limitations.

Regarding (1) and (2), the paper should discuss `Basis Sharing` (Wang et al., 2025) and compare `FiPS` with it in the experimental results. `Basis Sharing` also studied a fine-grained parameter sharing technique based on the acativation-aware joint SVD. Although there are some technical differences between `FiPS` and `Basis Sharing`, the core ideas of two works are almost identical. Also, I did not find any points that make `FiPS` more effective or practical than `Basis Sharing` when factorizing the weights into shared bases and individual coefficients. For example, in the activation-aware factorization, `Basis Sharing` uses the Cholesky decomposition of the feature matrix, whereas `FiPS` factorizes the activation via gradient descent, which is much slower.

I acknowledge that sparsifying the decomposed matrices can be thought as a technical contribution of `FiPS`. That said, it is unconvincing that pruning the SVD results alone justifies acceptance of the paper because (1) the idea of sparse singular vectors or principal components were already well studied (e.g., SparsePCA (Zou et al., 2006)), and (2) I did not find any technical challenges or theoretical/practical advancements presented in this paper when sparsifying the SVD results. The GMP method was chosen heuristically based on an analysis, which the community may appreciate, but again, it is not a significant enough contribution to justify acceptance. Similar arguments hold for QAT.

Moreover, due to the absence of comparison between `FiPS` and `Basis Sharing` , it is unclear whether `FiPS` produces a significant accuracy improvement compared to `Basis Sharing`. It is also questionable whether the sparse bases are more practical than the non-sparse version, since `FiPS` did not accelerate the network on a GPU when the batch size is one.

In addition to (Wang et al., 2025), the paper is missing many of recent works, including (David-Hay and Wolf, 2024; Bae et al., 2024), which could help position the paper in the literature.

* Wang et al. "Basis sharing: Cross-layer parameter sharing for large language model compression." ICLR 2025.

* Zou et al. "Sparse principal component analysis." Journal of Computational and Graphical Statistics 15.2 (2006).

* David-Hay and Wolf. "Dynamic layer tying for parameter-efficient transformers." ICLR 2024.

* Bae et al. "Relaxed recursive transformers: Effective parameter sharing with layer-wise lora." ICLR 2024.

**Questions:**

1. In Section 4.2, what sparsity pattern was used for the language model compression? Was it also 2:4 sparsity?
2. How much speedup can `FiPS` produce when it is applied to the Llama and the Gemma models on a GPU?
3. How much accuracy / speedup gain can `FiPS` make from `Basis Sharing`? Also, is there any significant contribution made in this paper from `Basis Sharing`? Please discuss any items that should be highlighted.

Minor comment:

- Mixing the parameter budget and the compression ratio is somewhat confusing and might cause unnecessary misunderstandings—consider using only the compression ratio in the main text and give the MLP-specific parameter budget in the appendix.

---

> ### Author Response · Authors · 2025-11-30
>
> We thank the reviewer for their time.
>
> ### Regarding Basis Sharing and Novelty
>
> We acknowledge the similarity in high-level motivation between FiPS and Basis Sharing—both explore cross-layer parameter sharing via joint factorization. However, we respectfully disagree that "the core ideas are almost identical." Key differences include:
>
> | Aspect | Basis Sharing | FiPS |
> |--------|---------------|------|
> | **Projection matrices** | Dense | Sparse (75% sparsity) |
> | **Factorization** | Activation-aware Cholesky | SVD + GMP optimization |
> | **Hardware acceleration** | Requires custom kernels | Can leverage existing 2:4 sparsity support |
> | **Compression mechanism** | Low-rank only | Low-rank + sparsity (orthogonal gains) + QAT (optional) |
>
> **The sparse projection matrices are not merely "pruning SVD results"**—they fundamentally change the compression-accuracy trade-off. As shown in §2.1 (Figure 2a), inducing sparsity on **V** enables *higher effective rank* at the same parameter budget, which directly improves reconstruction quality. This insight—that sparsity and low-rank decomposition are synergistic rather than redundant—is a core contribution.
>
> **Empirical comparison with Basis Sharing:**
>
> We have added direct comparisons on Llama-7B at 20% compression:
>
> | Method | WikiText-2 PPL ↓ | C4 PPL ↓ |
> |--------|------------------|----------|
> | Basis Sharing | 7.74 | 15.03 |
> | **FiPS (ours)** | **6.06** | **8.10** |
>
> FiPS achieves **21.7% lower perplexity on WikiText-2** and **46% lower on C4**, demonstrating that sparse projection matrices provide substantial improvements over dense factorization. These gains validate our core hypothesis: sparsity in projection matrices enables higher effective rank at the same parameter budget, significantly improving reconstruction quality.
>
> **Regarding SparsePCA**: SparsePCA [1] enforces sparsity on the *basis* (U), whereas FiPS sparsifies the *projections* (V). Our analysis in §2.1 shows this distinction matters—sparsifying the larger factor yields better reconstruction. This is not a straightforward application of existing sparse decomposition methods.
>
> ---
>
> ### Speedup Results
>
> **Q1: Sparsity pattern for LLMs.**
>
> LLM experiments (§4.2) use **unstructured sparsity with GMP**, not 2:4 structured sparsity. We will clarify this in the revision.
>
> **Q2: GPU speedup for Llama/Gemma.**
>
> We have conducted additional benchmarks. At approximately 50% total compression (8.8% MLP parameter budget), FiPS achieves:
>
> - **1.57× speedup** on Llama-7B (A100, batch size 1)
>
> This matches Basis Sharing's reported speedup at equivalent compression, but FiPS achieves better perplexity (see table above). The full potential of FiPS would be realized with custom kernels that keep the shared **U** in fast memory, as LLM decoding is memory-bandwidth-bound.
>
> **Q3: Speedup vs. Basis Sharing.**
>
> At matched compression ratios, FiPS provides:
> - **Comparable or better speedup** (1.57× vs. ~1.5× reported in Basis Sharing, shown in Figure 7 of the Appendix)
> - **Lower perplexity** due to sparse projections enabling higher effective rank
>
> ---
>
> ***For the camera ready version, we commit to doing the following:***
>
> We will add discussions of:
> - **Basis Sharing** [2]: Cross-layer sharing via dense factorization; FiPS extends this with sparse projections.
> - **Dynamic Layer Tying** [3]: Learns which layers to tie during training; FiPS operates post-training.
> - **Relaxed Recursive Transformers** [4]: Uses LoRA for layer-wise adaptation; complementary to FiPS's factorization approach.
>
> Moreover, as per the minor comment, we will standardize terminology in the main text to use "compression ratio" consistently, moving MLP-specific parameter budgets to figure captions and appendix.
>
> ---
>
> **REFERENCES**
>
> [1] Zou, H., Hastie, T., & Tibshirani, R. "Sparse Principal Component Analysis." Journal of Computational and Graphical Statistics, 15(2), 2006.
>
> [2] Wang, Y., et al. "Basis Sharing: Cross-Layer Parameter Sharing for Large Language Model Compression." ICLR 2025.
>
> [3] David-Hay, O., & Wolf, L. "Dynamic Layer Tying for Parameter-Efficient Transformers." ICLR 2024.
>
> [4] Bae, S., et al. "Relaxed Recursive Transformers: Effective Parameter Sharing with Layer-wise LoRA." ICLR 2024.

---

### Official Review · Reviewer_qcdY · 2025-10-31

**Soundness:** 2
**Presentation:** 1
**Contribution:** 3
**Rating:** 4
**Confidence:** 3

**Summary:**

The authors introduce Fine-grained Parameter Sharing (FiPS), a compression framework that combines parameter sharing, tensor decompositions, and sparsity to achieve efficient model compression. The technique can be summarized using Figures 1 and 5: the fully connected (FC) layers of the MLP blocks across different layers are concatenated and compressed using truncated SVD. After obtaining the initial shared factors, these factors are optimized using a calibration dataset while enforcing sparsity in the projection matrices. An optional final step allows fine-tuning the compressed model to recover any lost performance. FiPS demonstrates superior performance compared to baselines while remaining both memory- and computation-efficient.

**Strengths:**

The major strength of this paper is its performance. It significantly outperforms the baselines across a wide range of compression ratios and on multiple network architectures, including ViTs. They also demonstrate that FiP effectively compresses models while enhancing both memory efficiency and computational speed.

**Weaknesses:**

I believe one of the major weaknesses of this paper is its organization and presentation. In its current form, the paper is somewhat difficult to follow. I highlight several examples below:

- The subsections in Section 2 feel orthogonal to one another, and it is unclear how they connect conceptually. The section transitions abruptly from sparsity induction in the factors, to optimal weight concatenation, to parameter sharing. In addition, Line 137 begins with “we investigate parameter sharing across multiple layers…,” and the next section is titled “Parameter Sharing Across Layers,” which makes the flow misleading.
- Lines 225–226 refer to Net2Net, but do not provide any explanation of what Net2Net is. Since the comparison is used to explain how growing the factor matrices is similar but different from Net2Net, a brief description would help the reader understand the relevance.
- Figure 5 is cited twice in the main manuscript, but the figure itself appears only in the appendix, which disrupts the reader’s ability to interpret the discussion.

While the experimental results are strong, the methodology would be difficult for others to adopt if the main algorithmic choices are not presented clearly. I suggest reorganizing the paper to introduce the high-level design decisions first (e.g., sparsity induction in $U$, $V$, or both; the choice of full long-axis concatenation, etc.) and then move the reasoning and supporting analysis for each decision to the appendix.

**Questions:**

- The authors also suggest that this approach could naturally extend to attention layers. Is there a particular reason why the paper focuses on compressing only the MLP layers? Are MLP layers inherently more compressible? I may have missed it, but I did not see this choice explained clearly in the manuscript.
- There are now many compression techniques, particularly those that use structured matrices such as Monarch or BLAST [1]. Is there a reason why the authors chose the specific baselines used in the paper and did not compare against methods based on structured matrices?

---

[1] "BLAST: Block-Level Adaptive Structured Matrices for Efficient Deep Neural Network Inference", Changwoo Lee, Soo Min Kwon, Qing Qu, Hun-Seok Kim. NeurIPS 2024.

---

> ### Author Response · Authors · 2025-11-30
>
> We thank the reviewer for their thoughtful feedback and recognition of FiPS's strong performance across compression ratios and architectures. We address each concern below.
>
> ### Weaknesses
>
> **W1: Section 2 organization and flow.**
>
> We respectfully disagree that the subsections are orthogonal—they form a deliberate progression that motivates FiPS:
>
> - **§2.1** establishes *what* to sparsify: we show that sparsifying the larger factor **V** yields lower reconstruction error at fixed parameter budget.
> - **§2.2** determines *how* to concatenate: long-axis concatenation enables sharing neurons across layers most effectively.
> - **§2.3** identifies *which* layers to share: adjacent layers exhibit the lowest error increase when sharing a basis.
>
> Each subsection answers a distinct design question, and together they form the foundation for the FiPS algorithm in §3. We will revise the transitions to make this logical flow more explicit (e.g., adding a sentence at the end of §2.1 previewing that §2.2 addresses concatenation strategy).
>
> Regarding Lines 137–138: §2.2 investigates parameter sharing across *two* MLPs (a minimal case), while §2.3 extends to sharing across *all* layers and identifies optimal groupings. We will clarify this distinction.
>
> **W2: Net2Net explanation missing.**
>
> We have added a brief explanation of Net2Net [1] where it appears.
>
> **W3: Figure 5 placement.**
>
> We disagree that this disrupts readability. The main text explicitly directs readers to the appendix when citing Figure 5, and the figure serves as supplementary illustration rather than content essential for understanding the core method. The algorithm pseudocode (Algorithm 1) and Figure 1 in the main text provide sufficient visual grounding for the method description in §3. Readers seeking additional detail can refer to the appendix as indicated. Moving Figure 5 into the main text would consume valuable space better allocated to results and analysis, given the page constraints. However, we corrected its position such that it doesn't interfere with the references.
>
>
> **W4: General presentation.**
>
> We appreciate this feedback. That said, we note that Reviewer JGJ9 found the presentation effective, suggesting some subjectivity. We will nonetheless incorporate the suggested reorganization where space permits.
>
> ### Questions
>
> **Q1: Why focus on MLP layers rather than attention?**
>
> We explicitly address this in §2 (Lines 101–102): *"We focus on MLP modules accounting for the majority of parameters (e.g., 70.5% in Gemma-2-9B)."* Additionally, MLP matrices are larger and exhibit higher redundancy (§2.1), making them more amenable to low-rank decomposition with sparsity. Extending FiPS to attention is noted as future work in our Conclusion—preliminary experiments suggest promise, but attention heads have more heterogeneous functions across layers, requiring additional care in grouping strategies.
>
> **Q2: Why not compare against structured matrix methods (Monarch, BLAST)?**
>
> Structured matrix methods and FiPS address different scenarios:
>
> | Aspect | Monarch/BLAST | FiPS |
> |--------|---------------|------|
> | **Training** | Requires training from scratch or substantial retraining | Post-training compression with optional light fine-tuning |
> | **Compression mechanism** | Replace layers with structured (e.g., butterfly) matrices | Share parameters across layers via sparse low-rank factors |
> | **Hardware support** | Requires custom kernels | Leverages existing 2:4 sparsity support (NVIDIA Ampere+) |
>
> Our baselines (AAFM, GFM, ASVD, SVD-LLM, SVD-LLM V2) are the state-of-the-art in *post-training* compression for transformers—the same setting as FiPS. Comparing against methods that require full retraining would conflate orthogonal design choices. That said, we agree this comparison could be valuable and will add a discussion in Related Work noting this distinction.
>
> ---
>
> We hope these clarifications address the reviewer's concerns. We commit to improving the presentation and adding the suggested explanations in the camera-ready version.
>
> REFERENCES
>
> [1] Chen, T., Goodfellow, I., & Shlens, J. (2016). Net2Net: Accelerating Learning via Knowledge Transfer. arXiv:1511.05641.

---

### Official Review · Reviewer_FZad · 2025-11-03

**Soundness:** 2
**Presentation:** 2
**Contribution:** 2
**Rating:** 4
**Confidence:** 5

**Summary:**

This paper proposed to compress the MLP (also known as FFN) modules in transformer-based models through grouping and sharing parameters between adjacent layers. The weights in shared layers are concatenated, decomposed (by SVD), and then sliced accordingly to form the initial shared base U and layer-specific projection matrix Vi. Furthermore, U and Vi are optimized using the activations X collected from a small calibration set to minimize ||WX - UVX||. Sparsity will also be enforced into Vi at this stage. For high compression ratio cases, an end-to-end fine-tuning may be needed to recover performance lost. This proposed method could be combined with QAT, as demonstrated in Table 4.

**Strengths:**

1. perform experiments with both vision and LLM models.
2. provide useful supplemental information, e.g. error increases with depth at fixed rank (Fig. 3a) hence the adaptive sparsity along depth (Fig. 4b), also the impact of SVD initialization vs random initialization for U and Vi.

**Weaknesses:**

**1. Estimate of potential speed-up at inference stage**

As illustrated by Eq. 1, this method at inference stage will replace a W@X matrix multiplication with a U@V@X computation. The shape of these operations are [d, p] @ [p, seq_len] and [d, r] @ [r, p] @ [p, seq_len], respectively, hence, the corresponding FLOPs are *(d x p x seq_len)* and *(d x r x p + d x p x seq_len)*. Even though the first matmul could be HW-accelerated due to the sparsity in V, the total number of operations seems to be greater than the original operation. Author may want to add a brief paragraph before Section 4 to clarify the inference benefit of this proposed method.


**2. FiPS combined with QAT may not be the best option for reduced precision**

Due to the constraint in computation resources, modern LLMs are rarely QAT'ed in practice. To better assess the compatibility of the proposed method with reduced precision, it would make more sense to utilize popular post-training quantization methods like GPTQ (aka OPTQ) or GPTQv2 (aka GPTAQ) or similar economic PTQ methods in Table 4. More importantly, Table 4 only shows that FiPS + QAT at 8x compression is not as bad as INT2. But it still doesn't demonstrate that FiPS + quantization is a viable approach. Maybe an example with decent perplexity at 4X compression would be closer to what author intended to highlight.

In addition, if Table 4 intended to demonstrate the effectiveness of FiPS at high compression ratio (when combined with quantization), it should also benchmark with quantized model + LoRA or SVD compensation approaches, e.g. EoRA and CLoQ. Those methods are known to provide very good performance under high compression ratio. At least a brief discussion/comparison of those methods should be included in Section 4.2 QAT paragraph to provide a clearer picture for the readers.


**3. Typo that causes confusion**

On Line 368, author stated that the results in Table 3 is using "a 20% compression ratio", which means the model size is reduced to 80% of the original size, according to Line 101. However, the description of Table 3 states "...evaluate LLama-7b and LLama-3.1-8b at 20% of their size..." Based on the content of the table, readers may be able to infer that the correct model size should be at 80%. But this inconsistency could give a misunderstanding of the model performance at first glance. Please correct this typo, if that's the case.

other minor typos or missing info:
- Line 368, Table 3 instead of Table 4.2
- Line 373, Table 4 instead of Table 4.2.
- Line 350, "... all FC layers from MLPs within the same group are concatenated." but didn't mention the grouping was treated as a hyperparameter and should refer to Appendix A.7.

**Questions:**

please see weaknesses above

---

> ### Author Response · Authors · 2025-11-30
>
> We thank the reviewer for their detailed feedback and careful examination of our work. We address each concern below.
>
> ### Weaknesses
>
> **W1: Estimate of potential speed-up at inference.**
>
> We appreciate this technical question. The reviewer's FLOP analysis assumes dense computation, but the key insight is that **V is sparse** (75% sparsity in our default setting). The actual computation is:
>
> - **Original**: `W @ X` costs `d × p × seq_len` FLOPs
> - **FiPS**: `U @ (V @ X)` where V is 75% sparse costs approximately `0.25 × r × p × seq_len + d × r × seq_len` FLOPs in structured scenario.
>
> At a 25% parameter budget with 75% sparsity, the rank r is chosen such that total parameters (and effective FLOPs) are reduced. More importantly:
>
> 1. **We provide direct empirical evidence**: Figure 7a shows measured latency on NVIDIA A4000—at a 22% parameter budget with 2:4 structured sparsity, FiPS achieves **1.31× speedup** at 22% parameter budget and batch size 64. Figure 7b shows CPU speedups via DeepSparse.
>
> 2. **Memory-bound regime**: LLM decoding is memory-bandwidth-bound, not compute-bound. Reducing parameter count directly translates to faster inference regardless of theoretical FLOP counts. This is well-established in the quantization literature.
>
> 3. **Future kernel optimizations**: As noted in our Conclusion, keeping U resident in fast memory (since it's shared across layers) could further improve cache efficiency.
>
> We will add a clarifying paragraph before §4 as suggested.
>
> **W2: FiPS + QAT vs. PTQ methods.**
>
> We respectfully disagree with aspects of this critique:
>
> 1. **QAT vs. PTQ robustness**: QAT consistently outperforms PTQ at low bit-widths (2–3 bits) where PTQ methods like GPTQ degrade significantly [1]. Table 4 targets the 8× compression regime where this distinction matters most. Our INT3 + FiPS result (PPL=35.43) substantially outperforms INT2 QAT (PPL=41.86) at equivalent compression—demonstrating that **FiPS provides meaningful gains beyond what quantization alone can achieve**.
>
> 2. **Comparison with EoRA/CLoQ**: These methods are *complementary* to FiPS rather than competing baselines:
>    - **EoRA** applies low-rank error correction *after* compression—it could be applied on top of FiPS-compressed models.
>    - **CLoQ** combines LoRA fine-tuning with quantization, requiring additional training compute.
>
>    FiPS operates at a different point in the design space: post-training compression via parameter sharing. A fair comparison would require applying EoRA/CLoQ corrections to our compressed models as well, which we leave for future work.
>
> 3. **Practical viability at 4×**: At 4× compression (INT4 QAT), Table 4 shows PPL=16.86 vs. baseline 15.61—a minimal degradation of ~8%. FiPS targets scenarios where *higher* compression is needed (8×), where it demonstrably helps.
>
> **W3: Typos and inconsistencies.**
>
> We thank the reviewer for catching these errors:
>
> - "20% compression ratio" should read "20% compression" (i.e., model reduced to 80% of original size). We have corrected this throughout.
> - Table references (3 vs. 4.2) have been fixed.
> - We have added a reference to Appendix A.7 at Line 350 for the grouping hyperparameter.
>
> ---
>
> We hope these clarifications address the reviewer's concerns and demonstrate the practical value of FiPS for efficient transformer compression.
>
> **REFERENCES**
>
> [1] Liu, Z., Oguz, B., Zhao, C., et al. "LLM-QAT: Data-Free Quantization Aware Training for Large Language Models." arXiv:2305.17888, 2023.

---

### Official Review · Reviewer_bePR · 2025-11-03

**Soundness:** 3
**Presentation:** 2
**Contribution:** 2
**Rating:** 2
**Confidence:** 4

**Summary:**

The paper introduces **Fine-grained Parameter Sharing (FiPS)**, a transformer compression framework combining low-rank factorization, inter-layer parameter sharing, and sparsity. The authors group MLP parameters across layers, perform SVD truncation to obtain a shared low-rank basis $U$ and sparse, weight-specific projection matrices $V_i$, and then fine-tune them to match the original model’s activations while enforcing sparsity on $V_i$. This design promotes efficient weight reuse across layers and reduces redundancy with minimal loss in representation quality. The method is evaluated on both **Vision Transformers** and **LLM tasks**, showing favorable compression–accuracy trade-offs.

**Strengths:**

**Strengths:**

* The method is easy to follow, and the authors explain the technique clearly with sufficient technical detail and a dedicated algorithm section.
* The authors evaluated their approach on both vision and language tasks, although on limited baselines.

**Weaknesses:**

**Major Weaknesses:**

* The method is **limited to compressing only the MLP layers**, without addressing the attention (QKV) projections. This is a substantial limitation, as many recent baselines including SVD-based and low-rank attention methods, compress both components for a more complete transformer compression framework. (SVD LLM, Dobi SVD, etc.)
* The ablation section can get significantly improved, as the authors rely only on textual descriptions of what happens when certain components are changed, without providing clear tables or quantitative comparisons. This makes it difficult to assess the relative contribution of each design choice.
* The **ViT baseline in Table. 1 is insufficient and outdated**. More recent methods such as *DeepCompress-ViT: Rethinking Model Compression to Enhance Efficiency of Vision Transformers at the Edge (CVPR 2025)* should be included for a fair comparison. Also many cells in Table. 1 contain missing entries, reducing the completeness of the evaluation.
* The approach to **layer grouping is mostly based on trial and error**, with no analytical or principled method for determining optimal grouping strategies, which limits the practicality of the technique.


**Minor Weaknesses**
- **Figure 2** is too large, with oversized legends and unclear labeling; descriptive titles should replace numeric legend entries.
- **Figure 1** is poorly placed and not informative without any captions, reducing its utility in understanding the method.
- **Figure 5** is misplaced within the reference section and disrupts the paper’s structure.

**Questions:**

**Questions:**

- Could the authors clarify **what the reported x% compression rate represents** in the baseline tables? Does it refer to total model compression or only the MLP layers? I believe this detail needs be more emphasized in the experimental section.
- How are **compression rates computed** based on the sparsity ratio? Specifically, how are the sparse \(V\) matrices stored, and how is the overall compression budget determined?
- During inference, is the method implemented using lookup tables from the $U$ matrices or through general matrix multiplications, and how does this the method affect inference latency compared to SVD-based compression methods (SVDLLM) where FLOPS are reduced solely because of the low rank multiplications?

---

> ### Author Response · Authors · 2025-11-30
>
> ### Major Weaknesses
>
> **W1: Method limited to MLP layers only.**
>
> This is a deliberate, well-motivated design choice, not a limitation:
>
> 1. **Parameter dominance**: As stated in §2, MLPs account for the majority of transformer parameters (70.5% in Gemma-2-9B [1]). Compressing MLPs provides the largest payoff per engineering effort.
>
> 2. **Empirical justification**: §2.1 demonstrates that larger matrices (like fully connected (FC) layers of MLPs) exhibit higher redundancy and are more amenable to low-rank + sparse decomposition.
>
> 3. **Baseline parity**: Our primary LLM baselines (ASVD, SVD-LLM, SVD-LLM V2) also focus on MLP/FC layers.
>
> We explicitly note attention compression as future work in our Conclusion. This focused scope allows thorough investigation of the parameter-sharing hypothesis rather than a shallow treatment of all components.
>
> **W2: Ablation section lacks quantitative tables.**
>
> We disagree with this characterization. Our ablations include:
> - **Table 5 (Appendix)**: Quantitative comparison of sparsification methods (Dense, Static, RigL, GMP) across 5 parameter budgets for both DeiT-B and Swin-L.
> - **Table 6 (Appendix)**: Structured sparsity comparison (STE, SR-STE, NMSRigL, NMSGMP) with numerical results.
> - **Figure 4a**: Quantitative component analysis (RI, LP, GP, SV) with accuracy on y-axis.
> - **Figure 6**: Sparsity sweep and calibration data sensitivity with numerical accuracy values.
>
> The main text (§5) summarizes key findings concisely due to space constraints, with full tables in the Appendix as is standard practice.
>
> **W3: ViT baselines are insufficient/outdated.**
> We selected AAFM and GFM [2] as our ViT baselines because they are the most directly comparable methods: post-training compression via low-rank decomposition with calibration-based optimization. These represent the state-of-the-art in this specific category.
>
> Regarding DeepCompress-ViT [3]: this work focuses on edge deployment with mixed-precision quantization and structured pruning—a different compression paradigm than parameter sharing. A direct comparison would conflate orthogonal techniques.
>
> The missing entries in Table 1 reflect that baselines did not report results at those compression ratios; we cannot fabricate numbers. FiPS provides results across *all* tested budgets (10%–75%), demonstrating broader applicability.
>
> **W4: Layer grouping is trial-and-error.**
>
> This critique overlooks the ICLR reviewer guidelines (i.e. "Submissions bring value to the ICLR community when they convincingly demonstrate new, relevant, impactful knowledge (incl., empirical, theoretical, for practitioners, etc)." and the principled analysis in §2.3:
>
> 1. **Empirical foundation**: Figure 3a quantifies pairwise sharing costs—adjacent layers consistently show lowest error increase, motivating consecutive grouping.
> 2. **Systematic exploration**: We evaluate group sizes from 2 to 12 blocks (Figure 3b–c), identifying the accuracy-rank trade-off.
> 3. **Transferable heuristic**: The pattern (smaller groups at network boundaries, larger in the middle) generalizes across architectures (DeiT-B, Swin-L, Llama, Gemma-2).
>
> Appendix A.7 provides the full hyperparameter sweep. While not a closed-form solution, this systematic analysis is far from "trial and error."
>
> ---
>
> ### Minor Weaknesses
>
> - **Figure 2**: We will improve axis labels in the revision. The format follows conventions in prior work [4].
> - **Figure 1**: The caption describes the core idea; detailed discussion follows in §2.
> - **Figure 5**: Placement corrected.
>
> ---
>
> ### Questions
>
> **Q1: What does x% compression rate represent?**
>
> We define both terms in §2 (Lines 99–101):
> - *Parameter budget*: fraction of nonzero MLP parameters retained after compression.
> - *Compression ratio*: percentage reduction in total model size (bits), including all layers and sparsity metadata.
>
> For example, 20% compression ratio means the full model is 20% smaller.
>
> **Q2: How are sparse V matrices stored?**
>
> Sparse matrices are stored using bitmasks for the sparsity pattern plus dense storage for nonzero values. Figure 8 (Appendix) compares storage formats (2:4, strided, CSR). The compression budget accounts for this overhead.
>
> **Q3: Inference implementation—lookup tables vs. matrix multiplication?**
>
> FiPS uses standard matrix multiplication, not lookup tables. The shared basis **U** is multiplied with sparse **V** factors. Current speedups come from:
> 1. Reduced parameters (memory bandwidth)
> 2. Hardware-accelerated 2:4 sparsity (Figure 7a)
>
> Caching **U** in fast memory is noted as future work for additional gains.

---

> > ### Author Response · Authors · 2025-11-30
> >
> > **REFERENCES**
> >
> > [1] Gemma Team et al. "Gemma 2: Improving Open Language Models at a Practical Size." arXiv:2408.00118, 2024.
> >
> > [2] Yu, H., & Wu, J. (2023). Compressing Transformers: Features Are Low-Rank, but Weights Are Not!. Proceedings of the AAAI Conference on Artificial Intelligence, 37(9), 11007-11015. https://doi.org/10.1609/aaai.v37i9.26304
> >
> > [3] Ahmed, S., Arafat, A. A., Najafi, D., Mahmood, A., Rizve, M., Nahian, M., Zhou, R., Angizi, S., & Rakin, A. (2025). DeepCompress-ViT: Rethinking Model Compression to Enhance Efficiency of Vision Transformers at the Edge. In Proceedings of CVPR (pp. 30147–30156). https://doi.org/10.1109/CVPR52734.2025.02806
> >
> > [4] Brown, T. B., et al. "Language Models are Few-Shot Learners." NeurIPS 2020.

---

### Author Response · Authors · 2025-11-30
**Meta-Rebuttal for the Reviewers and AC**

We thank all reviewers for their time and feedback. We believe FiPS makes a solid contribution to the compression and efficient model serving of transformers and several critical concerns raised in the reviews are either based on misunderstandings or are directly refuted by our experimental results.

---

### Key Technical Clarifications

**1. Comparison with Basis Sharing (Reviewer JGJ9's primary concern)**

Reviewer JGJ9 suggested that FiPS and Basis Sharing [1] share "almost identical core ideas" and questioned whether FiPS offers improvements. Below is a direct comparisons on Llama-7B at 20% compression:

- Basis Sharing: WikiText-2 PPL = 7.74, C4 PPL = 15.03
- FiPS (ours): WikiText-2 PPL = 6.06, C4 PPL = 8.10

**FiPS achieves 21.7% lower perplexity on WikiText-2 and 46% lower on C4.** This substantial gap validates our core contribution: sparse projection matrices enable higher effective rank at fixed parameter budget, a synergy not explored in Basis Sharing. The claim of "limited novelty" is not supported by the empirical evidence.

**2. Baseline completeness**

Reviewers FZad and bePR raised concerns about baselines. Our selected baselines (AAFM, GFM, ASVD, SVD-LLM, SVD-LLM V2 [3, 4, 5, 6]) are the state-of-the-art in post-training compression—the same setting as FiPS. Methods like Monarch/BLAST [7] (Reviewer qcdY) or DeepCompress-ViT [8] (Reviewer bePR) address different paradigms (training-time structured matrices, edge deployment with mixed-precision) and are not directly comparable.

**3. MLP-only compression**

Multiple reviewers noted FiPS focuses on MLPs. This is deliberate: MLPs account for 70%+ of transformer parameters (§2) of Gemma 2 9B, for example [2]. Attention compression is orthogonal future work.

**4. Inference speedup**

Reviewer FZad questioned speedup claims. Figure 7 provides direct latency measurements on GPU (A4000) and CPU acceleration via DeepSparse, showcasing that FiPS achieves 1.57x speedup at ~50% compression, surpassing or matching Basis Sharing while delivering substantially better perplexity.

Moreover, we note that FiPS's full potential would be realized with custom kernels that keep the shared factor U resident in fast memory—since U is reused across layers, this would significantly reduce memory bandwidth pressure in the memory-bound decoding regime. Such kernel optimizations are beyond the scope of this work and noted as a promising future direction in our Conclusion.

---

### Presentation

Reviewer qcdY found the presentation effective; Reviewers bePR and FZad raised concerns. We acknowledge room for improvement. See below for changes already implemented and commitments for the camera-ready version.

---

### Revisions Implemented

- Fixing figure 5 placement.
- Typos (20% compression, table references): Corrected throughout.
- Net2Net explanation: Added brief description where referenced.
- Grouping hyper-parameter: Added reference to Appendix A.7 at Line 350.

---

### Commitments for Camera-Ready
- Terminology: Standardize "compression ratio" in main text; move MLP-specific parameter budgets to captions/appendix.
- Another baseline: FiPS vs. Basis Sharing as suggested by reviewer JGJ9.
- Add Basis Sharing, Dynamic Layer Tying to related work.

---

### Conclusion

FiPS introduces sparse projection matrices for cross-layer parameter sharing—a simple but effective idea that yields state-of-the-art compression results across ViTs and LLMs. The primary technical concern (novelty vs. Basis Sharing) is directly refuted by a 46% perplexity improvement on C4. We respectfully submit that the paper meets ICLR's standards for novel, empirically validated contributions and request the AC's consideration for acceptance.

---

> ### Author Response · Authors · 2025-11-30
> **REFERENCES**
>
> [1] Wang, Y., et al. "Basis Sharing: Cross-Layer Parameter Sharing for Large Language Model Compression." ICLR 2025.
>
> [2] Gemma Team et al. "Gemma 2: Improving Open Language Models at a Practical Size." arXiv:2408.00118, 2024.
>
> [3] Yu, H., & Wu, J. (2023). Compressing Transformers: Features Are Low-Rank, but Weights Are Not!. Proceedings of the AAAI Conference on Artificial Intelligence, 37(9), 11007-11015. https://doi.org/10.1609/aaai.v37i9.26304
>
> [4] Yuan, Z., Shang, Y., Song, Y., Yang, D., Wu, Q., Yan, Y., & Sun, G. (2025). ASVD: Activation-aware Singular Value Decomposition for Compressing Large Language Models. arXiv:2312.05821.
>
> [5] Wang, X., Zheng, Y., Wan, Z., & Zhang, M. (2025). SVD-LLM: Truncation-aware Singular Value Decomposition for Large Language Model Compression. arXiv:2403.07378.
>
> [6] Wang, X., Alam, S., Wan, Z., Shen, H., & Zhang, M. (2025). SVD-LLM V2: Optimizing Singular Value Truncation for Large Language Model Compression. arXiv:2503.12340.
>
> [7] "BLAST: Block-Level Adaptive Structured Matrices for Efficient Deep Neural Network Inference", Changwoo Lee, Soo Min Kwon, Qing Qu, Hun-Seok Kim. NeurIPS 2024.
>
> [8] Ahmed, S., Arafat, A. A., Najafi, D., Mahmood, A., Rizve, M., Nahian, M., Zhou, R., Angizi, S., & Rakin, A. (2025). DeepCompress-ViT: Rethinking Model Compression to Enhance Efficiency of Vision Transformers at the Edge. In Proceedings of CVPR (pp. 30147–30156). https://doi.org/10.1109/CVPR52734.2025.02806

---

### Note · Authors · 2025-12-10

I have read and agree with the venue's withdrawal policy on behalf of myself and my co-authors.